# Research on the Coordinated Governance Mechanism of Cross-Regional and Cross-Basin Ecological Compensation in the Yangtze River Delta

**DOI:** 10.3390/ijerph19169881

**Published:** 2022-08-11

**Authors:** Zhen Yu, Qingjian Zhao

**Affiliations:** College of Economics and Management, Nanjing Forestry University, Nanjing 210037, China

**Keywords:** Yangtze River Delta, cross-regional, ecological compensation, coordinated governance mechanism, institutional analysis and development framework, social network analysis

## Abstract

The development of a regionally integrated economy promotes the development of river basin ecological compensation toward cross-regional coordinated governance. The ecological compensation in the Yangtze River Delta has developed by leaps and bounds, which is conducive to the research on the collaborative governance mechanism. Taking the ecological compensation policy data in the Yangtze River Delta as the research object, and using the social network analysis method, this paper analyzes the current situation of cross-basin cooperation in the Yangtze River Delta. A collaborative governance network is formed with 74 ecological compensation agreements, and the distribution law of the overall collaborative network is found. Using IAD to decompose the ecological compensation agreement rules, 303 institutional units were obtained, of which, 198 were selection rules, accounting for 65%. The research results show that: (1) The ecological compensation cooperation in the Yangtze River Delta region is mainly concentrated in the Jiaxing Jiashan, Wujiang District, Suzhou, and Qingpu District, Shanghai, forming a close cooperation triangle network, and Shanghai plays a strong “intermediary” role in it. (2) In institutional grammar analysis, the formulation of rules is biased toward choice rules and payoff rules, that is to say, the content of the rules is mostly how to cooperate and how to reward and punish but there is a lack of specific action scenarios and standards. The combination of social network and institutional analysis and development framework is conducive to the study of the ecological collaborative governance mechanism of the Yangtze River Delta, breaking the gap between different fields and regions, enhancing the enthusiasm for multi-subject governance in the Yangtze River Delta region, and giving full play to the effectiveness of multi-subject governance.

## 1. Introduction

China’s rapid economic development has made the contradiction between development and the environment increasingly obvious. The problems of environmental pollution and ecological damage are increasing day by day, seriously hindering the sustainable and healthy development of China’s economy and society. In order to achieve new progress in ecological civilization construction during the “14th Five-Year Plan” period, it is necessary to speed up the construction of the ecological civilization system and build a beautiful new China. The ecological compensation mechanism is of great significance in promoting the common protection and governance of the ecological environment of the river basin, the coordinated development of high regional quality, and the realization of the value of ecological products, and is an inherent demand of an ecological civilization [1]. With the development of human society, the pollution of the ecological environment is becoming more and more serious. According to the United Nations “Water Resources Development Report”, the global rivers, lakes, seas, and the freshwater resources on which human beings depend have deteriorated sharply, and more than half of the 500 rivers are on the verge of depletion or pollution [2]; in addition, the ecosystem service value will also be affected [3,4]. People are also aware of the importance of the coordinated development of social development and the ecological environment. In the process of promoting the modernization of the national governance system and governance capacity, collaborative governance has become an indispensable concept and means of green integrated development. In the face of increasingly complex development problems, constrained by factors such as capabilities, resources, responsibilities, goals, etc., a single local government has been unable to solve multi-dimensional problems such as environmental protection, public health, etc. that involve political, economic, and social spatial externalities [5].

Chinese scholars have formed relatively rich results in the research on collaborative governance. Cross-domain ecological environment collaborative governance is faced with governance dilemmas such as insufficient environmental protection cohesion, difficulty in defining responsibility sharing, and difficulty in reaching a cooperation consensus [6]. Improved collaborative governance mechanisms, such as decision-making and prevention, participation in implementation, communication and coordination, information sharing and mutual trust, performance evaluation and supervision, and ecological compensation mechanisms, ensure that cross-domain ecological and environmental governance can achieve practical results [7]. By cultivating a community of shared interests and responsibilities and reshaping the trust mechanism between governance subjects, local governments, enterprises, and the public can form an organic synergy and realize the “synergy” of regional ecological and environmental governance [8]. It is necessary to seek a dynamic balance in the political game between various governance subjects, and it is necessary to build an effective commitment and cooperation mechanism, reputation mechanism, information communication mechanism, incentive mechanism, and supervision mechanism [9]. From the perspective of governance, the central and local government’s choice of four governance strategies, namely, bureaucratic outsourcing collaboration, adaptive adjustment collaboration, market contract collaboration, and multi-participatory collaboration, affect the collaborative governance of the ecological environment. In terms of governance longitude, the achievement of ecological environment collaborative governance is affected by the learning path between the cooperating subjects, that is, whether environmental protection policies have undergone local government innovations and pilots, feedback, and selection of environmental protection policies, and that environmental protection policies have been upgraded to regulations. The system waited for three stages and finally survived [10]. As an emerging governance model, collaborative governance meets the requirements of ecological environment governance and can play the role of coordinating all parties and balancing values to help promote the common understanding and internal legitimacy of stakeholders, and embedding collaborative governance in the cross-domain ecological environment. The problem-solving process has necessity and possibility [11].

As the birthplace of collaborative governance theory in the West, due to different political systems and other factors, the research on ecological collaborative governance is different from that in China. Through the negotiation mechanism, a coordination organization for the interests of multiple subjects is established; and, based on coordinating the relationship between the subjects, an integrated river basin agency conducts comprehensive governance across river basins [12]. Contreras proposed that public participation, democratic development, and restraint on power are important factors for the coordinated governance of the river basin’s ecological environment [13]. Lockwood pointed out that absorbing as many subjects as possible to participate in collaborative governance will help the structure and function of the governance subjects correspond to the scope of influence of ecological and environmental problems and make the governance measures of ecological and environmental problems more targeted and effective [14]. Heijden proposed that ecological collaborative governance not only requires a collaborative mechanism but also cannot be separated from the constraints of systems and regulations [15]. Erickson, after studying the case of river basin ecological environment governance in the United States, believes that collaborative governance is continuously carried out for different goals. Through the continuous accumulation of specific and subtle goals, the importance of the overall river basin ecological environment system can be further discovered, and collaborative governance can be promoted. The goals and means of the system focus on the protective utilization of the overall ecological environment system [16].

The in-depth study of ecological collaborative governance is conducive to the coordinated development of the ecological environment and economy. The proposal of regional concepts such as “economic belt”, “economic zone”, and ecological compensation will involve different degrees of cooperation between different regions. The collaborative governance mechanism is conducive to resolving the conflict over the core interests of the utilization of upstream and downstream water resources involved in cross-regional cooperation. Existing research has not fully answered the questions in the cross-border ecological collaborative governance mechanism and system formulation. In view of this, this paper takes the cross-border collaborative governance institutional documents in different regions of the Yangtze River Delta as the research object, taking the cooperation network as the starting point, and macroscopically describing the current status of collaborative governance in the Yangtze River Delta. Combined with micro-institutional analysis, the deficiencies in the institutional documents of collaborative governance are found.

## 2. Study Area and Data Source

As shown in the blue area in Figure 1, the study area is where the Yangtze River Delta region is located. Some cities involved in the study and the locations of related rivers are marked in the figure. The Yangtze River Delta region has always been characterized by strong economic vitality, a high level of development, an excellent ecological environment, and a profound historical and cultural heritage. It is an important strategic part of China’s socialist modernization construction. As of the end of 2019, the Yangtze River Delta had a population of 227 million and an area of 358,000 square kilometers. The urbanization rate of the permanent population exceeds 60%, which is less than 4% of the country’s land area, creating nearly one-quarter of China’s total economic output and one-third of China’s total import and export volume. The Yangtze River Delta region itself has unique and advantageous geographical conditions, with riverside and seaside, vertical and horizontal rivers, excellent natural endowments, a good ecological environment, better environmental protection level and effectiveness than the whole country, and a solid economic foundation.

Before the implementation of the ecological compensation system in the Yangtze River Delta, the Yangtze River Delta paid attention to the research on the ecological compensation mechanism in different fields, including the ecological compensation of the river basin. Therefore, the Yangtze River Delta region is a region with a relatively high level of social and economic development in China, and the ecological and environmental protection systems and policies are relatively complete. As of March 2020, there have been 10 cross-provincial river basin ecological compensation cases implemented in the country [17]. Under the new round of ecological compensation system reform, the three provinces and one city in the Yangtze River Delta have achieved good results. Therefore, the research object of this paper is the Yangtze River Delta, which is easier to understand and study.

In terms of ecological compensation, the Yangtze River Delta region is actively exploring ecological protection compensation mechanisms in various fields including river basin ecological compensation. Currently, the three provinces and one city in the Yangtze River Delta region are cooperating in various aspects of ecological integration. The cooperation agreement signed in governance and ecological compensation shows that the three provinces and one city in the Yangtze River Delta region are making efforts and are in close cooperation with environmental and ecological improvement. Demonstration areas for the integrated development of ecological and green development have been formed, mainly in Suzhou Wujiang, Zhejiang Jiashan, and Shanghai Qingpu. At the same time, other regions have also formed their own exclusive cooperation areas. The implementation of ecological compensation is inseparable from the cooperation among various government departments. With the continuous emergence of cross-regional public issues and public affairs, the traditional administrative mode of administrative regions will gradually transition to regional administration and regional governance models [18].

The relevant ecological compensation agreements in the article mainly come from the policy documents published on the government websites of all parties, as well as the agreements involved in the articles about the ecological environment on authoritative news websites. In addition, some experts were consulted, and a total of 74 multi-party agreements and plans for cross-border ecological governance and ecological compensation were collected.

## 3. Research Methods

As a geographical unit under the administrative directive, the Yangtze River Delta region cooperates in tourism, agriculture, labor, and other fields in order to better promote the integration process. In the specific implementation of different fields, there is no specific administrative instruction on how to carry out activities, which requires rational choices between different regions. This forms a network of cooperation in nesting and cooperation zones. The cooperative network is a voluntary relationship structure between members. In cross-regional ecological collaborative governance, the cooperative network is a voluntary relationship structure formed between members. Using the method of the social network to study the synergy mechanism, it is easier to find the current status and problems of ecological synergy governance. Based on the content of the specific agreement, IAD links the rules with specific elements, which is conducive to distinguishing and analyzing the functions that each rule must undertake in the realistic situation of the ecological collaborative governance of the Yangtze River Delta.

### 3.1. Social Network Analysis (SNA)

The social network analysis method is a common method to study the network structure, so this paper will use the ecological compensation agreement as the medium and use the network density, node centrality, and cohesive subgroup indicators in SNA to analyze the network characteristics [19]. The network density reflects the degree of density of nodes in the network. After binarizing the agreement signing quantity table, the closer the calculated network density is to 1, the more closely all nodes in the network are connected [19]. The expression formula of network density *D_i_* is shown in Formula (1):(1)$$D_{i}=\frac{2L}{NN-1}$$
among them, *N* is the city involved in the network and *L* is the actual number of connections.

Node centrality is used to measure the centrality of nodes in the network, mainly involving degree centrality and betweenness centrality [19]. Degree centrality is used to describe the importance of nodes in the network and the connection ability in the network. In directed networks, degree centrality can be divided into point-out degree and point-in degree. The click-out degree is used to measure the cross-border cooperation activities carried out by the node, and the click-in degree is used to measure the node’s acceptance of cross-border cooperation [19]. In terms of agreements, in-degree and out-degree are the same; after all, signing an agreement is a two-way process. Assuming that the out-degree and in-degree are a and b, which can be calculated by Ucinet software (Version 6.689, Stephen Borgatti, UC Irvine, Irvine, CA, USA), and the number of network nodes is *N*, the expression formula of the relative degree centrality *C_Di_* is shown in Formula (2):(2)$$C_{D\dot{l}}=\frac{a+b}{N-1}$$

Betweenness centrality represents the frequency of a node that appears on the shortest path between two other nodes and can also be understood as the number of all shortest paths in the network passing through this node [19]. Betweenness centrality is used to probe the pivot points in the network that perform the function of “intermediary” or “transit”. The formula for calculating relative betweenness centrality is shown in Formula (3):(3)$$C_{i}=\frac{2C_{j}}{\left(n-1)(n-2\right)}$$
among them: *C_i_* represents relative centrality, *C_j_* represents absolute centrality, and its calculation formula is Formula (4):(4)$$C_{\dot{J}}=\sum_{j}^{n}\sum_{k}^{n}b_{jk}(i),j\neq k\neq i,\;j<k$$

Assuming that there are $g_{jk}$ paths between point *j* and point *k*, the number of paths between points *j* and *k* passing through point *i* is represented by $g_{jk}\left(i\right),$ and the ability of point *i* to control the communication between these two points is represented by $b_{jk}i$; its calculation formula is Formula (5)
(5)bjk(i)=gjk(i)gjk

### 3.2. Institutional Analysis and Development Framework (IAD)

One of the challenges of applying institutional theory to a policy setting is translating key concepts into reliable strategies that can be observed. The Institutional Analysis and Development (IAD) framework defines institutions as “shared concepts that humans use in repetitive contexts organized by rules, norms, and policies” [20]. The framework was first proposed by the Ostroms in 1982 and has been continuously developed since then and widely used in the analysis of different practical situations. As a theoretical analysis framework, three groups of exogenous variables, such as natural material conditions, community attributes, and application rules, as well as action situations and results, are the core elements of the institutional analysis and development framework [21].

As shown in the framework diagram in Figure 2, the application rules are important external variables that affect the action situations and actors in the actor arena, are the basic determinants of the formation of social incentive structures [22], and are also an effective way to guide actor situations and actors [23].

Ostrom summarized the application into seven rules: position, boundary, choice, scope, aggregation, information, and payoff rules [24]. The cumulative effect of these seven rules affects different aspects of the action scenario, and each rule type affects the structure of the action scenario. The relevant rules are described according to Table 1. Position rules determine the positions of participants [23]; boundary rules affect the number of participants, their attributes and resources, whether they are free to enter, and the conditions for leaving [20]; choice rules assign roles on a particular node, i.e., a collection of actions that must or cannot be taken and are linked to outcomes [23]; scope rules define potential outcomes that may be affected and limit actions associated with specific outcomes [24]; aggregation rules affect when actors choose actions, i.e., the level of control performed [23]; information rules affect participants’ knowledge information sets [24]; and payoff rules affect the benefits and costs assigned to specific combinations of actions and outcomes, thereby determining the motivation and deterrence of actions [24].

## 4. Research Result

### 4.1. Cross-Regional and Cross-Basin Ecological Compensation Collaborative Governance Network Analysis

Nowadays, it is unrealistic to rely on a single government to solve multidimensional problems with spatial externalities such as environmental protection, public health, etc. that involve political, economic, and social aspects [25]. How to promote high-quality development on the basis of ecological environmental protection, realize regionally coordinated, integrated development without breaking administrative divisions, effectively build a demonstration area for ecological and green integrated development in the Yangtze River Delta and create an ecologically friendly integration, is a challenge that we are currently facing. The cities involved in the current study overlap in space, such as the Suzhou and Suzhou Wujiang Districts. However, because the parties to the agreement are different, it cannot be concluded that the agreement signed by the Wujiang District government is Suzhou because Wujiang District is within Suzhou. It is signed by the city government, so we must ignore the geographical factor and discuss it separately.

Figure 3 below shows the statistics based on the collective agreements. Since 2012, cooperation on ecological collaborative governance has appeared in the Yangtze River Delta. The number of agreements on ecological collaborative governance, and ecological compensation in the agreement, is increasing. There is also a growing emphasis on the importance of synergy between regions. From 1 agreement in 2012 to 20 in 2021, it can be seen that it is difficult to effectively promote ecological governance by relying on a single government at present. It is necessary to rely on cooperation and coordination between different departments in different regions to further promote the process of ecological green integration.

#### 4.1.1. Cross-Regional Ecological Collaborative Governance Network

Gephi was used to build the overall ecological collaborative governance cooperation network in the Yangtze River Delta region, as shown in Figure 4 below, with different colored lines representing the cooperation network between different regions, and the thickness of the lines indicating the degree of cooperation between different regions. The size of the label indicates that the region has a high degree of centrality in the overall cooperation network, that is, it can participate more in cooperation in different regions in the ecological collaborative governance.

Combined with Ucinet, the centrality analysis of the ecological cooperation network in the Yangtze River Delta was carried out. The specific results are shown in Table 2. It can be found that:(1)The larger the value of the degree centrality, the greater the direct influence of this node on other nodes, the greater the degree centrality of the node, and the higher the influence in the network. The top three cities are Qingpu District in Shanghai, Jiashan County, Jiaxing City, Zhejiang Province, and Wujiang District, Suzhou. It can be seen that the cooperation between the three is close, and the Qingpu District of Shanghai is in the absolute center, which has an impact on Jiashan County, Jiaxing City, and Wujiang District, Suzhou.(2)The greater the betweenness centrality, the stronger the role the city plays in the collaboration with other cities in the overall collaboration network. Table 2 shows that Shanghai Qingpu District, Zhejiang Jiaxing, and Suzhou Wujiang are synergistic in the Yangtze River Delta, an important role in governance. Qingpu District in Shanghai serves as the center of divergence and aggregation of the network, as well as the center of economic and green technology development. It has strong dominance and control over the flow of resources, such as capital and technology, and acts as an “intermediary”. It shows that Shanghai Qingpu District pays attention to the protection of the ecological environment, and will connect the surrounding cities, driving their actions in ecological compensation and working together to protect the ecological environment.(3)A higher closeness centrality indicates that, in a network of collaborative governance, it can be associated with other cities more quickly and cannot easily be controlled by other cities, playing the role of “central actor” in the spatial association network. In Table 2, Shanghai Qingpu, Zhejiang Jiashan, and Suzhou Wujiang are important fulcrums to support the ecological collaborative governance network in the Yangtze River Delta. The cooperation between the three places can also effectively promote the ecological compensation cooperation of surrounding cities, thereby expanding the governance network.

#### 4.1.2. Cross-Basin Ecological Collaborative Governance Network

Ecological collaborative governance in the Yangtze River Delta will involve different basins, and different basins will involve different cooperation networks. Therefore, the six basins of Dianshan Lake, Taipu River, Taihu Lake, Xin’an River, Yangtze River, and Huaihe River are used as the research points to analyze the social network characteristics of its collaborative governance. It should be noted that the cities involved in each river basin are only listed according to the collective agreements. Some cities do not involve cooperation, which does not mean that no ecological efforts have been made, they are only listed in the network diagram. It is generally in the position of an isolated point, so we will not discuss it for a single node.

The cooperation between the Xin’an River Basin, the Yangtze River Basin, and the Huai River Basin is mainly between Anhui and Jiangsu, while the Dianshan Lake Basin, Taipu River Basin, and Taihu Lake Basin are mostly concentrated in Jiangsu, Zhejiang, and Shanghai. Although the Yangtze River Delta needs to promote the development of green integration, it can be seen from Table 3 that different basins have formed their own subgroups.

Gephi was used to make a network map of various basins, as shown in Figure 5 and Figure 6.

It can be seen from Figure 5 in combination with Table 3 that the cooperation is denser in a and b. Most of the cities between the two collaborative governance and cooperation between the two are the same. There are intersections in ecological governance. The overall cooperation network of the basin is not large, mainly based on Shanghai Qingpu, Wujiang, Suzhou, and Jiashan, Zhejiang.

In Figure 6, in terms of a, b, c, and d, the gap between the cooperation in the four basins is large. a and b have the center points of their own cooperation network but involved a small city, which has formed a small child group. Additionally, c and d involve many cities but the frequency of cooperation between cities is low, so it is difficult to form a good cooperation network such as Figure 5a,b.

The centrality numerical analysis was carried out according to Ucinet, as shown in Table 4.

(1)In the analysis of the degree center, we find that there are cities that play an important role in different river basins, such as Shanghai Qingpu District in the Dianshan Lake Basin, Huzhou City in the Taihu Lake Basin, Jixi County in the Xin’an River Basin, and so on. In the vast majority of cities with the highest degree of centrality in the watershed, there will be a huge gap between the degree of centrality of other cities, indicating that the city has a far-reaching influence on the integrated green development of the watershed and will give relevant green integrated development from a macro perspective. However, this also shows that there are problems such as a lack of system and imperfect supporting policies in the ecological collaborative governance of the Yangtze River Delta, which is not conducive to the horizontal and vertical development of the collaborative network in multiple regions.(2)The greater the betweenness centrality, the stronger the role the city plays in the collaboration with other cities in the overall collaboration network, and the greater the role of bridge and intermediary in the network. Cities with high betweenness centrality have the strongest ability to control policy resources, and are also the intermediaries and coordinators in the policy network. Similar to the degree centrality, the difference between the cities with the largest betweenness centrality and other cities in different watersheds is too large, so it is difficult to involve more cross-regional, cross-departmental, and different-level subjects in the collaborative governance of cross-watersheds in green integrated development.(3)Cities with close centrality and high intermediary centrality highly overlap. These cities have high policy resource control capabilities and are often at the core of the policy network. They can maintain close contact with other departments and are not easily affected by other departments’ sectoral impact. However, comparing the lower betweenness centrality of other cities in the overall network shows that Shanghai Qingpu District, Huzhou City, Jixi County, and other cities that play an important role in this watershed, in the overall network, form a cluster of sub-network alienation and overall network.

It can be seen that the nodes that play an important role in the coordinated ecological governance of the river basin are the same as the nodes that play an important role in the overall ecological coordinated governance of the Yangtze River Delta. The entire Yangtze River Delta is dominated by the Dianshan Lake and Taipu River basins. The cooperation is mainly carried out around the three cities of Qingpu District, Jiashan County, and Wujiang District. Cities in different basins also have different degrees of collaborative governance cooperation with these three cities.

### 4.2. Analysis of Ecological Collaborative Governance Rules

In order to better understand the status quo of ecological collaborative governance in the Yangtze River Delta, we focus on the social networks formed by the three places that play a leading role in the social network analysis: Shanghai Qingpu District, Suzhou Wujiang District, and Zhejiang Jiashan, and extract some ecological compensation agreements to analyze the content characteristics. 

As shown in Table 5 below, from some of the agreements collected around Qingpu, Wujiang, and Jiashan, we can find that Qingpu, Wujiang, and Jiashan carry out coordinated cross-border ecological governance around the Dianshan Lake Basin and the Taipu River Basin. The main content is to establish the “joint river chief system”, and pay attention to the sharing of monitoring information so as to effectively carry out activities.

Although many measures for collaborative governance are given in the agreement, emergencies are also explained. However, no specific compensation and incentive measures were given. This shows that the cooperative network mainly formed by the three places of Qing, Wu, and Shan place emphasis on the prevention of ecological environment protection and does not involve punitive measures for adverse consequences.

According to the rules of the IAD system analysis, combined with the specific content in the system documents, different observation units in the rules are selected and classified according to seven classification requirements. As shown in Table 6, under the current ecological collaborative governance, the rules of different institutional documents will correspond to seven kinds of rules.

Therefore, we combined the IAD rules to count the distribution of the above-mentioned rules in the Yangtze River Delta in the agreement. As shown in Figure 7 below, a total of 74 agreements involve a total of 303 institutional units, most of which are based on job choice rules. The content of the agreement mostly involves assigning a set of actions that a role on a specific node can, must, or cannot take, and establishes a link with the result. While other aspects of the rules are rarely involved, the focus is on ecological environmental protection, and there is a lack of punitive measures for adverse consequences.

The lack of specific compensation and incentive measures in the ecological compensation agreement for cross-border cooperation is mainly because the rivers or watersheds managed by different regions are different, so the compensation standards and compensation amounts are different. The compensation method is to adopt the government ecological compensation, that is, each government pays a certain amount, and the compensation is made according to whether the measurement results of the water quality of the section are qualified or not. Because of the existence of boundary rules, it is difficult to form cross-border cooperation between different administrative regions, and it is difficult to continue to expand the cooperation network of collaborative governance.

In addition, the overall ecological compensation does not involve provisions on compensation willingness, because of the lack of aggregation rules and information rules, and no good group decision-making. In the end, the ecological compensation is to make up for the loss of the masses in production and life. If the analysis of the willingness to pay and the willingness to pay for ecological compensation is not carried out, the compensation amount formulated may not play a good compensatory role. A good compensation mechanism will need to be formed.

The main problem encountered in cross-border ecological compensation is not just signing an agreement, but how to carry out more comprehensive and coordinated cooperation, so that cooperation can be carried out under different governance levels and rule frameworks, in order to improve the efficiency of ecological compensation in the basin [32]. The formulation of the agreement only depends on the selection and payment rules and cannot achieve the sustainable development of ecological compensation, so it is necessary to consider the impact of several other rules on the effectiveness of ecological compensation.

As shown in Table 7, each rule has certain problems. At present, most ecological compensations are paid based on water quality results rather than conservation behaviors [33] and rely on indicators related to biodiversity and ecosystem services. Position rules are the main body of ecological governance and cross-border ecological cooperation, but in actual operation, the planning of positions is relatively simple. Under the government-led management model, it is easy to lead to the lack of multi-subject “governance”. The diversification of position rules can promote the formation of a mature management system. Therefore, government departments can delegate power to local governments and introduce some supervisors, non-governmental organizations, etc., which can effectively improve the efficient implementation of ecological compensation policies. Boundary rules are the least involved because they are about the entry and exit “conditions” of the agreement, so this is related to the scale effect of the overall cooperation and should be effectively combined with market means (such as contract auctions or bidding) so that more people can participate in it; it is also conducive to solving the problem of insufficient funds for compensation projects in ecological compensation. Aggregation and information rules are one category; aggregation rules focus on participation in decision-making, while information rules focus on the disclosure of relevant information during implementation, which is a way for the public to know, understand, and obtain information. Therefore, attention should be paid to the establishment of aggregation rules and information rules so that more people can participate in decision-making and improve the participation of local citizens. Citizens can participate in the discussion and understand the current situation in order to make effective suggestions and improve everyone’s enthusiasm for participation.

## 5. Discussion

Through the analysis of the cooperation network and system, we found that the current Yangtze River Delta has certain defects in both the collaborative governance mechanism and related support systems; the collaboration mechanism and system are complementary. Because of the imperfect coordination mechanism, effective cooperation and communication cannot be achieved, and the system agreement cannot be fully discussed when it is formulated so there is no strong constraint. Similarly, the imperfect system is not conducive to the horizontal and vertical development of the cooperation network. It will only form sub-networks with strong cooperation in some places and cannot promote the development of the overall network.

As the so-called “stones from other mountains can attack jade”, at present, there are relatively mature cases of cross-regional collaborative governance at home and abroad worthy of study and reference. In the Tennessee River Basin of the United States, in order to effectively solve the problems of environment and resource utilization, the Tennessee River Basin Administration was established. Its main responsibility is to formulate a series of specific goals for the development and construction of the river basin and includes the active participation and coordination of multiple responsible subjects, such as local governments, enterprises, and the public. Similarly, in Australia’s Murray–Darling Basin, in addition to the Basin Ministers Council, which is a government agency, there is also a Basin Council, a non-governmental organization that is mainly responsible for the implementation of specific plans and coordinating the planning and management of the basin. It can be seen that an effective coordination mechanism is inseparable from the cooperation of multiple subjects. In addition, the Thames River Basin in the United Kingdom has promulgated a complete set of laws and regulations, making detailed regulations on the subject, object, and content of accountability for the ecological environment, and clarified the goals of ecological responsibility at each level through the legal system. Under the comprehensive institutional guarantee of the Guangdong-Hong Kong-Macao Greater Bay Area, different regions carry out horizontal and vertical cooperation, internal negotiation and governance, and effectively manage the ecological environment. In contrast, in the cooperation network of the Yangtze River Delta, Shanghai occupies an absolute leading role and the main body of its cooperation is only Wujiang District and Jiashan County at the government level; there is no other non-government or public participation. This can explain why, in the analysis of the system, the selection rules are dominant and there is a lack of supervision and communication. Therefore, the formulation of the system is only to achieve the goal, there is no specific standard, and there is a lack of information exchange. In the end, the cooperation network can only be limited to a small area.

The determination and decomposition of ecological responsibility goals are the dynamic operation links of cross-administrative ecological environment collaborative governance. Table 8 summarizes the collaborative governance experience at home and abroad. In a word, Under the perfect ecological environment collaborative governance system, multi-subject participation and horizontal and vertical communication and negotiation governance can ensure the sustainable and healthy development of the Yangtze River Delta ecological collaborative governance towards a broader cooperation network.

## 6. Conclusions

For the research on the ecological collaborative governance mechanism of the Yangtze River Delta, the combination of SNA and IAD can effectively study the current collaborative governance mechanism of the Yangtze River Delta and the problems existing in the existing system from both macro and micro perspectives. The study found:①Cross-border collaborative governance in the overall Yangtze River Delta watershed; Shanghai is absolutely dominant, and the areas where cooperation is concentrated are limited to three places: Qingpu District, Jiashan County, and Wujiang District. The analysis of the cooperation network in other different watersheds shows that the network characteristics are similar to the overall network characteristics of the Yangtze River Delta watershed and the formed sub-networks of clusters are alienated from the overall network in which they are located.②Whether analyzing the agreements in Qingpu District, Jiashan County, and Wujiang District from a partial perspective, or analyzing the Yangtze River Delta Agreement from an overall perspective, the rules of the agreement are mainly based on selection rules. It can be seen that in cooperation between different regions, too much emphasis is placed on doing things, while ignoring the standards of doing things.

Finally, it should be mentioned that this paper also has shortcomings. First of all, it does not propose specific improvement measures for the cooperation network formed by the Yangtze River Delta but only analyzes its current cooperation status and characteristics. Secondly, institutional analysis can further analyze the content of institutional agreements in a more extensive and specific manner. This article only exemplifies a few agreements which have certain limitations.

## Figures and Tables

**Figure 1 ijerph-19-09881-f001:**
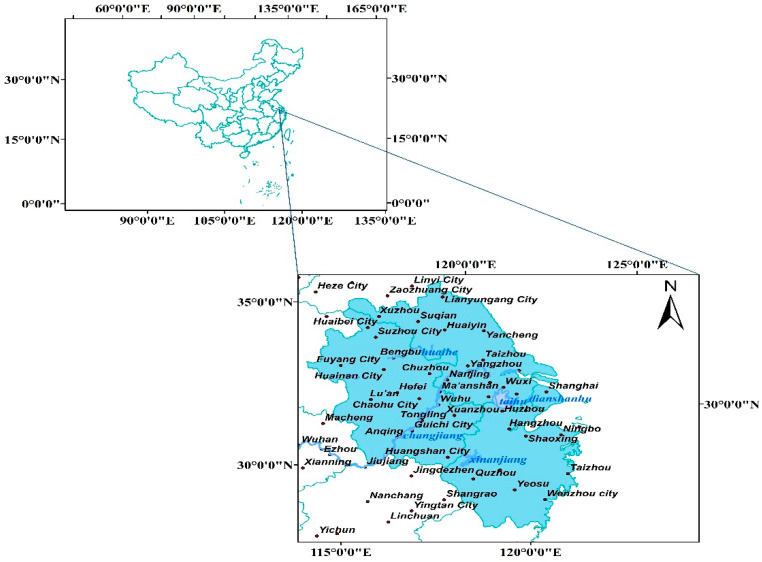
Yangtze River Delta region.

**Figure 2 ijerph-19-09881-f002:**
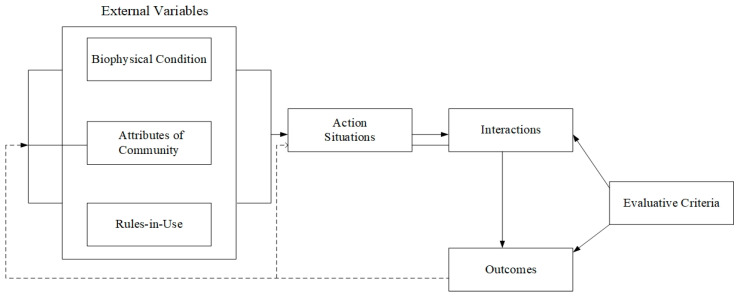
Institutional Analysis and Development (IAD) Framework.

**Figure 3 ijerph-19-09881-f003:**
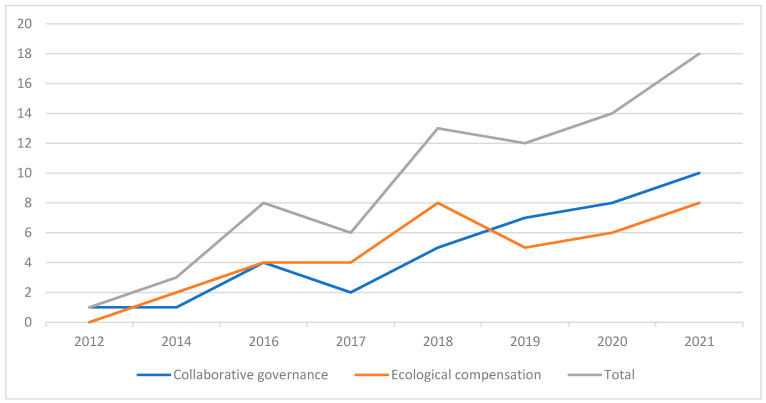
Trends in the number of signed cross-border ecological collaborative governance agreements in the Yangtze River Delta.

**Figure 4 ijerph-19-09881-f004:**
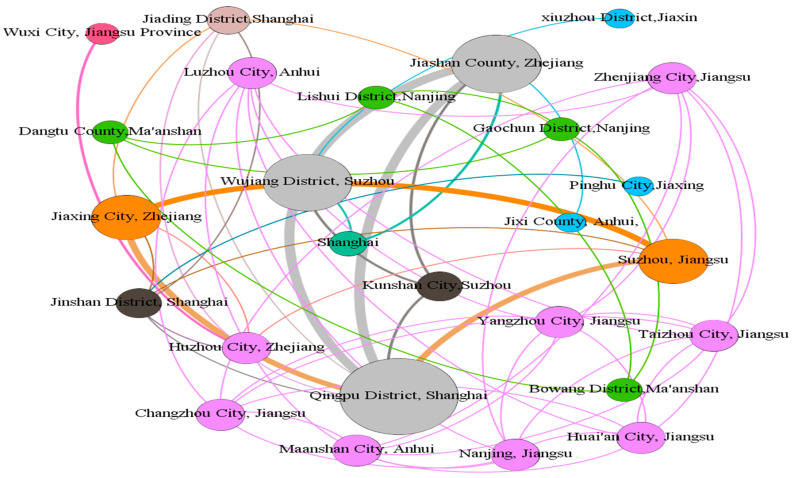
Cross-regional ecological collaborative governance network in the Yangtze River Delta.

**Figure 5 ijerph-19-09881-f005:**
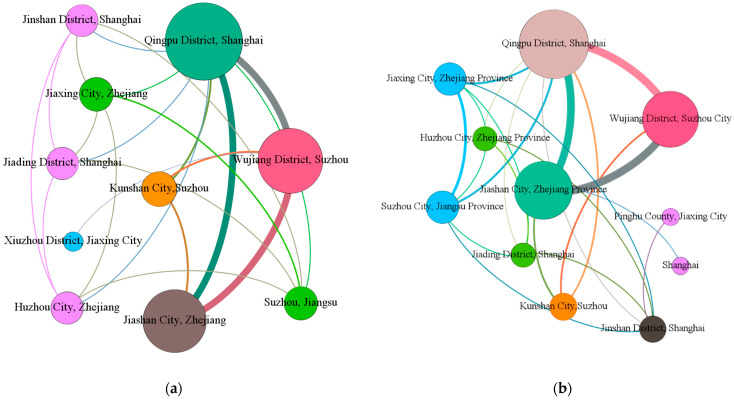
(**a**) Dianshan Lake Basin and (**b**) the Taipu River Basin.

**Figure 6 ijerph-19-09881-f006:**
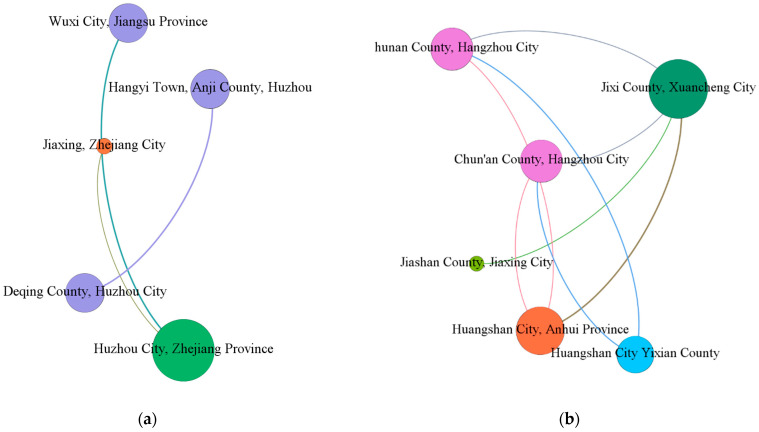
(**a**) Taihu Basin, (**b**) Xin’an River Basin, (**c**) Yangtze River Basin, and (**d**) Huaihe River Basin.

**Figure 7 ijerph-19-09881-f007:**
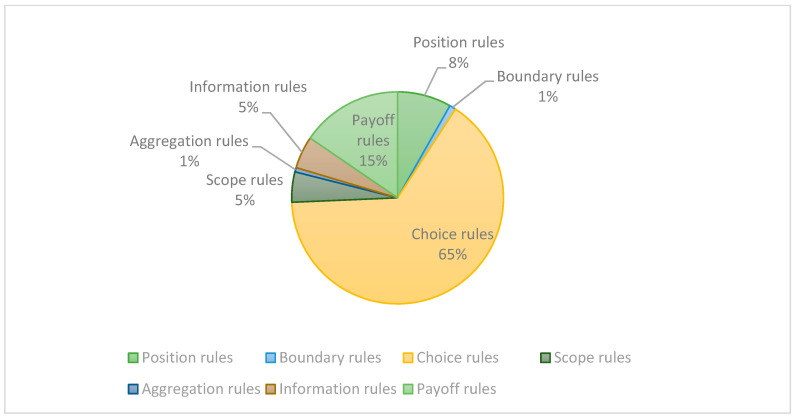
The proportion of rules in cross-border collaborative governance.

**Table 1 ijerph-19-09881-t001:** Rule types and descriptions of the IAD framework.

IAD Framework Rules	Description Keywords	Explanation
Position rules	“Be”/“Become”	Determine the role position
Boundary rules	“Enter”/“Leaving”	Participant’s attributes and enter/leave conditions
Choice rules	“Do”	Must or must not be unable to act
Scope rules	“Occur”	Potential result
Aggregation rules	“Combined effect”	How to achieve group decision-making
Information rules	“Send”/“Receiving”	Knowledge information set
Payoff rules	“Payment”/“get”	Income cost

**Table 2 ijerph-19-09881-t002:** Centrality analysis of the spatial correlation network of ecological collaborative governance in the Yangtze River Delta.

City	Degree Centrality		Betweenness Centrality		Closeness Centrality	
	Centrality	Rank	Centrality	Rank	Centrality	Rank
Qingpu District, Shanghai	29.0	1	84	1	0.32	1
Jiashan County, Zhejiang Province	20.0	2	33	4	0.30	2
Wujiang District, Suzhou City	19.0	3	33	3	0.29	3
Jiaxing City, Zhejiang Province	14.0	4	36	2	0.25	5
Suzhou City	12.0	5	6	7	0.28	4
Jinshan District, Shanghai	6.0	7	24	5	0.20	7
Huzhou City, Zhejiang Province	7.0	6	24	6	0.22	6

**Table 3 ijerph-19-09881-t003:** Cities involved in each basin of the Yangtze River Delta.

Basin	Number of Cities	Surrounding Cities
Dianshan Lake Basin	10	Qingpu District, Shanghai Wujiang District, Suzhou, Jiashan County, and Zhejiang
Taipu River Basin	11	Qingpu District, Shanghai Wujiang District, Suzhou, Jiashan County, and Zhejiang
Taihu Basin	5	Huzhou City and Zhejiang Province
Xin’anjiang Basin	6	Jixi County, Xuancheng City, Chun’an County, and Hangzhou City
Yangtze River Basin	15	Maanshan and Anhui
Huaihe River Basin	15	Huai’an, Jiangsu, Cuzhou, and Anhui

**Table 4 ijerph-19-09881-t004:** Analysis of the centrality of each watershed.

Basin	City	Degree Centrality		Betweenness Centrality		Closeness Centrality	
		Centrality	Rank	Centrality	Rank	Centrality	Rank
Dianshan Lake	Qingpu District, Shanghai	21	1	35	1	0.75	1
	Wujiang District, Suzhou City	16	3	8	2	0.6	2
	Jiashan City, Zhejiang Province	17	2	0	3	0.53	3
	Suzhou Kunshan	6	4	0	3	0.53	3
Taipu River Basin	Qingpu District, Shanghai	24	1	48	1	0.63	1
	Wujiang District, Suzhou City	20	3	0	4	0.45	3
	Jiashan County, Jiaxing City	21	2	18	2	0.48	2
	Jiaxing City, Zhejiang Province	9	4	10	3	0.63	1
Taihu Basin	Huzhou City, Zhejiang Province	3	1	1	1	0.5	1
	Wuxi City, Jiangsu Province	2	2	0	2	0.44	2
	Hangyi Town, Anji County	2	2	0	2	0.4	3
Xin’anjiang Basin	Jixi County, Xuancheng City	5	1	2	1	0.56	1
	Huangshan City, Anhui Province	4	2	0	2	0.5	2
	Huangshan City Yixian County	3	3	0	2	0.38	4
	Jiashan County, Jiaxing City	1	3	0	2	0.45	3
Huaihe River Basin	Nanjing Lishui District	10	1	6	1	0.44	2
	Ma’anshan, Anhui Province	6	2	0	2	0.47	1
	Liuhe District, Nanjing City	2	3	0	2	0.35	3
Huaihe River Basin	Huai’an City, Jiangsu Province	12	1	12	1	0.54	1
	Luzhou City, Anhui Province	6	2	0	2	0.5	2
	Liu’an City, Anhui Province	4	3	0	2	0.35	3

**Table 5 ijerph-19-09881-t005:** Analysis of the contents of some ecological compensation agreements.

Protocol Name	Involves Cities	Agreement Type	Basin Involved in the Agreement	Main Content
“List of docking matters for the integration of Qing Kun Wu Shan in the strategic coordination area around Dianshan Lake (2018–2020)”	Qingpu, Kunshan, Wujiang, and Jiashan	Cross-border ecological collaborative governance	Dianshan Lake Basin and Tai Pu River Basin	There are 11 actions in the ecological environment, including the establishment of a joint river chief system, an environmental monitoring network, etc.
“Qingkun Wushan Water Area Cleaning Integration Collaboration Framework Agreement”	Qingpu, Kunshan, Wujiang, and Jiashan	Cross-border ecosystem governance	Dianshan Lake Basin and Tai Pu River Basin	In order to eliminate the harmful effects of water hyacinth, Taipu River and other rivers and lakes have established a linkage salvage mechanism.
“2019 Integrated Development Work Plan for Shanghai Qingpu, Jiangsu Wujiang, and Zhejiang Jiashan”	Qingpu, Wujiang, and Jiashan	Cross-border ecological collaborative governance	Dianshan Lake Basin and Tai Pu River Basin	National demonstration zones for ecological civilization construction are carried out in adjacent areas, and emergency plans and response measures for joint prevention and control of heavy pollution in the Yangtze River Delta region are compiled.
“Cooperation Framework Agreement on Integrated Ecological Environment Comprehensive Governance”	Qingpu, Wujiang, and Jiashan	Cross-border ecological collaborative governance	Dianshan Lake Basin and Tai Pu River Basin	Discussion on cooperation and sharing of monitoring information in the integrated demonstration area.
“Framework of in-depth cooperation agreement on the upstream and downstream of Taipu River”	Qingpu, Wujiang, and Jiashan	Cross-border ecological collaborative governance	Taipu River Basin	Through co-construction, protection, and management, the coordinated governance of Taipu River water resource protection is realized.
“Work Plan of Inter-Provincial Collaboration Mechanism for Taipu River Water Resources Protection”	Qingpu, Wujiang, and Jiashan	Cross-border ecological collaborative governance	Taipu River Basin	Unified and shared monitoring data to promote real-time early warning of the regional water ecological environment.
“Work plan for joint response to abnormal water quality indicators in transboundary sections of the Taipu River Basin”	Qingpu, Wujiang, and Jiashan	Cross-border ecological collaborative governance	Taipu River Basin	Attention is paid to emergency linkage mechanisms for emergencies.
“Framework Agreement on Inter-provincial Cooperation on Taipu River Drinking Water Sources and Jiashan Emergency Water Source Cooperation”	Shanghai and Jiashan County	Cross-border ecological collaborative governance	Taipu River Basin	The two sides jointly carry out the protection of the Taipu River water source.
“Framework of in-depth cooperation agreement between upstream and downstream management units of Taipu River”	Jiashan County Taipu River Management Office, Shanghai Huangpu River upstream embankment (pump gate) Management Office, and Tai Puhe Engineering Management Office of Wujiang District, Suzhou City, Jiangsu Province	Cross-border ecological collaborative governance	Taipu River Basin	The three places strengthen daily linkage management and overall coordination in project management and construction.
“Water Quality and Quantity Monitoring Data Exchange and Sharing Protocol”	Suzhou, Jiaxing, and Qingpu	Cross-border ecological collaborative governance	Dianshan Lake Basin and Tai Pu River Basin	The real-time data of the automatic water quality measurement station of the Jinze section of the mainstream of the Taipu River and the data of the Taipu Gate water automatic measurement station are shared.
“Implementation plan for joint environmental prevention and control in Qingpu, Jiashan and Wujiang”	Qingpu, Jiashan, and Wujiang	Cross-border ecological collaborative governance	Dianshan Lake Basin and Tai Pu River Basin	An inter-provincial regional environmental supervision, pollution prevention, and emergency response linkage working mechanism is established.
“Inter-provincial Cooperation Mechanism for Water Resources Protection of Taipu River-Water Quality Early Warning Linkage Scheme (Trial)”	Qingpu, Wujiang, and Jiashan	Cross-border ecological collaborative governance	Taipu River Basin	Qingpu, Wujiang, and Jiashan will share information, strengthen mutual communication, and establish and improve a joint operation mechanism for the ecological environment monitoring, supervision, and emergency response in the three places.
“Joint working mechanism for water environment cleaning, joint prevention and governance in the junction area”	Suzhou City and Jiaxing City	Cross-border ecological collaborative governance	Dianshan Lake Lake	Established a system of joint pollution control in border areas, joint and cross law enforcement in different places, and rectification and tracking within a time limit.

**Table 6 ijerph-19-09881-t006:** Classification of governance rules based on IAD.

Institutional File	Observation Unit	Rule Classification
“Memorandum of Cooperation on Implementing Credit Joint Rewards and Punishments in the Field of Environmental Protection in the Yangtze River Delta (including specific measures)” [26]	Credit rewards and punishments in the environmental protection field in the Yangtze River Delta region are responsible for the formulation of annual plan formulation, daily communication, overall coordination, etc.	Position rules
“Environmental monitoring linkage work plan for the Yangtze River Delta ecological and green integrated development demonstration area” [27]	The sources of drinking water in Tai Puhe County, Jiashan County, in the depth of Shanghai Qingpu and Wujiang, Jiangsu.	Boundary rules
“Outline of the Yangtze River Delta Regional Integrated Development Plan” [28]	The Tai Pu River and other rivers and lakes have established a linkage salvage mechanism to jointly ensure the environmental appearance of the waters and eliminate the impact of the hazards of water hyacinth.	Choice rules
“The overall plan of the Yangtze River Delta ecological and green integrated development demonstration area” [29]	Jointly carry out the construction of the ecological restoration belt of Taihu Binhu to enhance the self-cleaning ability of Taihu water body.	Scope rules
“Guiding Opinions on Establishing a Joint Prevention and Control Mechanism for Sudden Water Pollution Events in the Upstream and Downstream of Inter-provincial River Basins” [30]	In accordance with the requirements of a unified credit code, unified catalog standards, and unified database, uniformly shared the exchange system and uniformly publicized the release system requirements to build an integrated “Credit Yangtze River Delta” platform.	Aggregation rules
“Memorandum of Cooperation on Implementing Credit Joint Re-wards and Punishments in the Field of Environmental Protection in the Yangtze River Delta (including specific measures)” [26]	Give full play to the role of various social subjects such as the industry association, credit service agency, and big data companies, and increased the collection of market credit information.	Information rules
“Compensation Agreement for Horizontal Ecological Protection of Qiantang River (Upstream) Watershed” [31]	When the average annual value of the water quality of the town (street) reaches or exceeds the target value, the upstream township (street) obtain ecological compensation.	Payoff rules

**Table 7 ijerph-19-09881-t007:** Analysis of specific problems.

Rule Name	Example	Defect	Improvement Method
Position rules	Credit rewards and punishments in the environmental protection field in the Yangtze River Delta region are responsible for the formulation of annual plan formulation, daily communication, overall coordination, etc.	The main body is relatively single and there is a lack of multiple subject “governance”	The introduction of mass supervision and non-governmental organizations to make effective cooperation so that the rules of the job diversify.
Boundary rules	The sources of drinking water in Tai Puhe County, Jiashan County, in the depth of Shanghai Qingpu and Wujiang, Jiangsu.	The agreement involves less and the boundary is blurred	Combined with market means, set up high exit barriers to enhance the long-term nature of ecological compensation projects.
Choice rules	Jointly carry out the construction of the ecological restoration belt of Taihu Binhu to enhance the self-cleaning ability of Taihu water body.	Lack of specific indicators	While clarifying the indicators, the problems within the scope of management should be solved carefully.
Scope rules	In accordance with the requirements of unified credit code, unified catalog standards, and unified database, uniformly shared the exchange system and uniformly publicized the release system requirements to build an integrated “Credit Yangtze River Delta” platform.	Lack of mass decisions	Form the decision-making method from top to bottom to improve the effectiveness of decision-making.
Aggregation rules	Give full play to the role of various social subjects such as the industry association, credit service agency, and big data companies, and increase the collection of market credit information.	Lack of free acquisition information	The government should hold meetings in time to publish information such as changes in funds and ecosystem services to the public.

**Table 8 ijerph-19-09881-t008:** Cross-domain collaborative governance cases at home and abroad.

Basin	Main City	Governance Participant	Governance System or Rules	Existing Problems	Solution
US Tennessee River Basin	Virginia, North Carolina, Georgia, Alabama, Tennessee, Kentucky and Mississi	The Tennessee River Basin Management Bureau, a non-governmental organization, and the public	“Tennessee River Basin Authority Law”	Environment and resource utilization issues	Relied on the governments, local governments, corporate public, and other subjects in the basin to participate in governance.
Murray-Darling Basin, Australia	New South Wales, Victoria, Queensland, South Australia	Comprehensive river basin management agencies including the Board of Board, the Basin Committee, and the Community Advisory Committee	“The Murray– Darling Basin Agreement”	Environmental management issues	Diversified negotiation and autonomy of the public sector and private sector.
Thames Valley, UK	British capital London and more than 10 cities along the river	Thames Water Management Bureau	“Water Law”	Environmental management issues	The regional level and the basin level have set up a unified basin management department at multiple levels.
Guangdong-Hong Kong-Macao Greater Bay Area	Guangdong, Hong Kong, Macau	The Guangdong-Hong Kong, Guangdong-Macao Environmental Cooperation Group and its subordinate task force	“Outline of the Development Plan for the Guangdong -Hong Kong- Macao Greater Bay Area”	Ecological environmental protection issues	Improved institutional settings and institutional guarantees, realized regional horizontal cooperation and internal consultation and governance.

## Data Availability

The data presented in this study are available on request.

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
