# Peer review of "Research on the Coordinated Governance Mechanism of Cross-Regional and Cross-Basin Ecological Compensation in the Yangtze River Delta"

_ijerph, 2022, doi:10.3390/ijerph19169881_

Round 1

Reviewer 1 Report

This study utilized the social network analysis method and Institutional Analysis and Development Framework to analyze the cross-basin cooperation, conduct institutional analysis, summarize the characteristics, and discover shortcomings of the current ecological compensation agreements in the Yangtze River Delta. The topic is interesting and valuable for the ecological protection and sustainability in the Yangtze River Delta. However, the manuscript requires the careful revision before it is publishable. The suggestions/comments for this paper are as follows:

(1) What are the advantages of this study compared with other previous researches? Please illustrate the advantage/ innovation of this study in the abstract, introduction and conclusion sections.

(2) Page 4, Line 159, the authors mentioned that “The relevant ecological compensation agreements in the article mainly come from the policy documents published on the government websites”. Please add the references of the website link of some representative published agreements.

(3) Figure 1 need to be revised. The map of China should not have the map scale. Please add the graticule (longitude-latitude grid) for the map. Meanwhile, please identify all the geographic name involved in this study on the map of the Yangtze River Delta.

(4) It is not easy to read the number of the agreements in different years because the figure used the 3D chart view. Please use the two-dimensional chart view.

(5) Some words (geographic name) in Figure 4 can not be read because the fonts of them are too small. In Figure 7, the readers also can not find the detailed proportion values of different rules in cross-border collaborative governance.

(6) Compared with Section 4, Section 5 is too short. It is better to provide more profound discussions of the involvement of non-government, enterprises, and the public. The 5.2 section only provide several analysis results and suggestions for ecological sustainability. This section should conclude the paper, summarize the main ideas, emphasize the advantages of the study and point out the inadequacies.

(7) There are still some syntax errors, capital errors and punctuation errors in the main text. Please carefully revise the manuscript.

Reviewer 2 Report

This is a very worthwhile paper,

my comments are in order of appearance and not importance.

L 98- 101. I read Heijden to be saying that one needs both collaborative mechanisms AND regulation or compulsion (this seems an important point)

Line 117, definition of IAD, This acronym is defined at line 217, may be omit the acronym at line 117?lines 138 and 140 (and elsewhere, the expression 'in my country' in English expression in a paper like this would normally be 'China'. It may be different in Chinese? change or not as you wish.

line 172, I suggest, delete comma and which, substitute with to form the singular of Form in this sentence, to ,,,read regions to form a cooperative.. 

line 189. define Di in formular (density?)

Table at L 396, Targe, should be Large in Taipu river framework

Line 412 413, I think you mean Prevention of ecological environmental 'damage' not protection ?

Table at Line 415 in Choice rules row, what are water 'gourds' ?

Table 8 "Existence' do you mean 'problem here?

Row 2 I think you mean 'Murray' Darling, (correct name) not Mozan

Last Row  column 3 what is Australian Environmental Protection Group, doing in a row concerning Guangdong etc? 

L 538 A 'framework' not frame rk

L540 Mispelling Governance (not Goveranance)

Other comments, I found many of the reference did not appear in Google searches, but except for one, as above ' Heijden' they were reported correctly

Round 2

Reviewer 1 Report

The authors have responded to most of my previous comments. Now there are only two comments needed to be proposed as follows:

(1) Comment 3 “Figure 1 needs to be revised. The map of China should not have the map scale. Please add the graticule (longitude-latitude grid) for the map. Meanwhile, please identify all the geographic name involved in this study on the map of the Yangtze River Delta.” Responding to this comment, only the graticule has been added. Please carefully revise the figure.

(2) There are still some syntax errors, punctuation errors in the main text, especially the new added sentences and paragraphs. Careful proof-read of the manuscript is recommended.
